# PMG : Personalized Multimodal Response Generation with Large Language Models

Submission Id: 1913*

## ABSTRACT

The emergence of large language models (LLMs) has revolutionized the capabilities of text comprehension and generation. While previous research has focused on multimodal understanding using LLMs, there is little work on personalized generation, particularly in the context of recommender systems. This paper proposes the first method for personalized multimodal response generation using LLMs, showcases its applications and validates its performance via an extensive experimental study on two datasets. The proposed method, Personalized Multimodal Generator (PMG for short) first converts user behaviors (e.g., clicks in recommender systems or conversations with a virtual assistant) into natural language to facilitate LLM understanding and extract user preference descriptions. Such user preferences are then fed into a generator, such as a multimodal LLM or diffusion model, to produce personalized responses. To capture user preferences comprehensively and accurately, we propose to let the LLM output a combination of explicit keywords and implicit embeddings to represent user preferences. Then the combination of keywords and embeddings are used as prompts to condition the generator. We optimize a weighted sum of the accuracy score and preference score so that the generated responses have a good balance between them. Compared to a baseline method without personalization, PMG has a significant improvement on personalization for up to 8% in terms of LPIPS while retaining the accuracy of generated responses.

This paper proposes a method for generating multimodal responses, which is **prevalent in web applications**.

## CCS CONCEPTS

• **Information systems** → **Personalization**; **Multimedia content creation**.

## KEYWORDS

Multimodal, Large Language Model, Response Generation, Personalization

## 1 INTRODUCTION

Large language models (LLMs) have shown impressive text comprehension and generation capabilities. Many studies have sought to expand LLMs into the realm of multimodal understanding, particularly in the domains of image [38] and audio [21]. By integrating LLMs with modal-specific generators such as diffusion models [14] or multimodal LLMs [23], they can also be employed for multimodal generation tasks.

This paper aims to integrate personalization into multimodal response generation using LLMs, and to our best knowledge no work has addressed this task. Personalization is essential for improving user experience, better meeting users' needs and greatly increasing e-commerce revenue. Figure 1 shows an example of a chat tool.

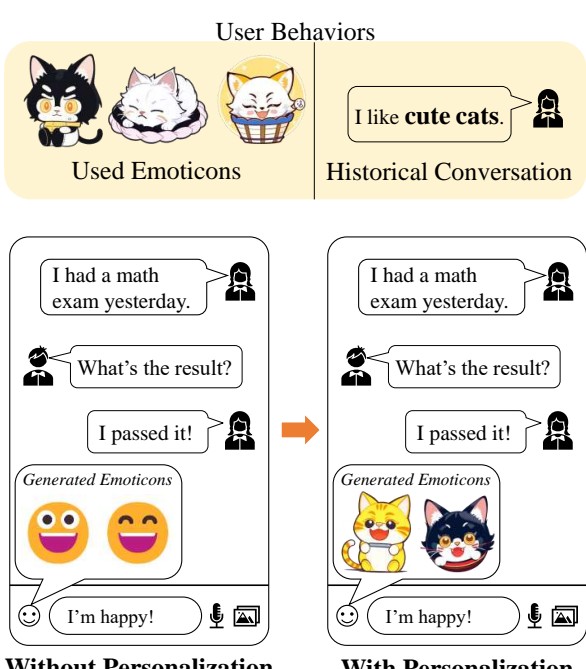

**Figure 1: The personalized generation based on user behaviors produces emoticons of a cute cat that are more appealing to cat lovers compared to the normal generation.**

When the user types in "I'm happy!", the chat tool understands the sentiment and provides emoticons of "happy". Popular apps such as TikTok, Discord, WeChat and Telegram already have functions similar to this, but they are without personalization, which is shown in the left part of Figure 1. After adding personalization, the chat tool would be able to generate personalized emoticons that are more appealing to the user as shown in the right part of Figure 1: based on the user's frequently used emoticons (cats in the example) or historical conversation ("I like cute cats" in the example), the chat tool would generate emoticons of happy cats.

As other example applications of multimodal generation, online advertisements need well-designed images of products to attract users. When recommending a movie, a personalized generator can produce personalized movie posters by amplifying the elements of a movie to the user's preference. Personalized clothing Apps can generate images of a person wearing a piece of clothing customized to her preferred height, weight, colors, etc. In video games, the generator can customize the background music to align with the content of the video and the user's preferred music genre. Moreover, as the generated responses reflect user preferences, they may be

leveraged as data augmentation to improve the accuracy of recommender systems.

In the above applications, we refer to the items we aim to generate without personalization as *target items*, e.g., the happy emoticons in the left part of Figure 1; note that there may be multiple target items. We refer to the items we aim to generate with personalization as *personalized target items*, e.g., the happy emoticons in the right part of Figure 1. The personalization process should make the candidate target items tuned to users' preferences while retaining their relevance to the candidate target items, which will be measured by an accuracy score in our experimental study. For example, if we generated a crying cat, then the accuracy score would be low in the example of Figure 1.

To address the aforementioned applications, we propose personalized multimodal response generation using LLMs (PMG for short). PMG first extracts a user's preferences from the user's behavior history, such as clicks in recommender systems or past conversations, and converts them to natural language such that they are easily understood by LLMs. The user preferences are then fed into a generator such as a multimodal LLM or diffusion model to produce personalized responses. There are still some challenges when implementing our method.

First, we find that merely representing user preferences as natural language, specifically keywords, may not be accurate because they have limited expressive ability whereas user preferences are abstract. To address this challenge, we propose to let the LLM output a combination of explicit keywords and implicit embeddings to represent user preferences. Then the combination of keywords and embeddings are used as prompts to condition the generator.

Second, conditioning the generation process also poses a challenge, as it requires accurately matching both the user preferences and a target item. Directly mixing these two factors may lead to an imbalance, potentially overshadowing one in the outcome. Therefore, our approach employs a weighted sum of the accuracy score and the preference score for each output response. The accuracy score measures the level of consistency with the target item, while the preference score gauges the degree of personalization. We optimize the sum by balancing the weights of the user preferences and target items, allowing us to address the imbalance and customize the degree of personalization.

Our contributions are summarized as follows:

- To our knowledge, this is the first work to address the problem of personalized multimodal generation using LLMs, and we demonstrate many potential applications.
- To address the problem, we propose a method named PMG, which first converts user behaviors into natural language so that LLMs can understand them and extract user preferences. Then the user preferences are fed into a generator to produce personalized responses.
- To address the challenge of capturing user preferences comprehensively and accurately, we propose to let the LLM output a combination of explicit keywords and implicit embeddings to represent user preferences, which are then used as prompts to condition LLMs for generating multimodal responses. We also propose to optimize a weighted

sum of the accuracy score and preference score so that the generated response has a good balance between them.
- An extensive experimental study validates the effectiveness of our method. Compared to a baseline method, which does not have personalization, PMG has significant improvement in personalization for up to 8% in terms of LPIPS while retaining the accuracy of generated responses.

## 2 RELATED WORK

### 2.1 Multimodal Generation

In the field of multimodal generation, previous research has investigated the utilization of generative models like Generative Adversarial Networks (GANs [11]) and Variational Autoencoders (VAEs [17]) to produce diverse and realistic outputs across various modalities. GANs employ a generator network and a discriminator network that undergo adversarial training. On the other hand, VAEs learn latent representations of data and generate new samples. Researchers have extensively explored and enhanced these approaches [4, 12].

The introduction of CLIP [25] revolutionized text-guided generation, making it more accessible. As a result, the diffusion model gained widespread popularity and became the method of choice for various generation tasks, including image generation [26] and audio generation [22]. It is often utilized as a downstream multimodal generator in LLM response generation. While most of these methods [24, 35] rely on natural language to establish the connection between the pre-trained LLM and generator, they are hampered by limited natural language expression capability. In contrast, TANGO [10] and GILL [18] employ informative hidden embeddings but are not stable and require substantial training to align their embedding space.

The current approaches to personalized generation, such as textual inversion [6] and DreamBooth [28], mainly focus on integrating new characters or image styles into a pre-trained diffusion model using multiple images. These approaches differ significantly from personalization based on user behaviors, which emphasizes the general interests of users rather than specific instances. Moreover, user behaviors encompass a combination of clicked items (including textual and visual features), conversations, and more, making it impractical to process using traditional personalized generation techniques.

### 2.2 LLM for Recommendation

Numerous studies have leveraged the exceptional reasoning capabilities of LLMs for recommender systems. The predominant approach involves converting historical click sequences and candidate pools into text and utilizing the language model to directly generate recommended items. Although it can yield favorable results even without training [7, 15, 30], this method lacks specific optimization for recommender tasks. In contrast, certain studies [1, 3, 31] adhere to this paradigm but employ techniques like prompt learning [33] or LoRA [16] for fine-tuning the language model and enhancing performance in recommender systems.

The aforementioned methods primarily utilize only the textual features of items, while P5 [8] takes a different approach by incorporating ID features. It utilizes special tokens representing user ID and

item ID within the language model. Building upon this, VIP5 [9] further enhances the recommendation process by adding item images, processed through a pre-trained image encoder, as visual tokens. However, a limitation of VIP5 is its reliance on the availability of original item images. If the original images are missing, it is unable to generate new images, restricting its applicability.

## 3 METHOD

### 3.1 Overview

Our proposed PMG is depicted in Figure 2. We leverage the reasoning abilities of a large language model to extract user preferences from historical behaviors (including clicks in recommender systems and conversations with a virtual assistant). The user behaviors are used to produce preference conditions, including explicit keywords in natural language (named preference keywords) by a frozen LLM and implicit embeddings (named soft preference embeddings) by a tunable bias correction LLM. Additionally, we also convert the target item into explicit keywords (named target item keywords) to serve as the target item conditions. Ultimately, the generator, which could be a diffusion model or multimodal LLM, produces the responses by incorporating and weighting personalized and target item conditions after the text encoder of the generator.

### 3.2 Generate Explicit Keywords

Given our objective of extracting user preferences using an LLM from behaviors, the simplest and most effective approach is to convert user behaviors into text and analyze them using the LLM. The generator typically has a limited input length (e.g., 77 tokens in Stable Diffusion [26]), making keyword summarization more informative than using full sentences. As a result, we design prompts for each scenario and leverage the zero-shot capability of the LLM without the need for training. In the following, we will discuss the process of prompt design.

*3.2.1* ***Preprocess of user behaviors.*** We consider two types of user behaviors: historical clicks $H = \{h_1, h_2, \cdots\}$ and conversations $C = \{c_1, c_2, \cdots\}$. The input features could be multimodal, including texts, images, audios, etc. Normally, the LLM has the ability to handle complex texts, so we can simply feed the texts into it. But the texts may be long (e.g., a plot synopsis of a movie), and concatenating all of them from an item sequence exceeds the token length limit of the LLM. In this case, we summarize the text features of each item and conversation into a short sentence using the LLM as preprocessing. For the other features, we convert them into text using a caption model (e.g. BLIP-2 [19], CLAP [5]) or using multimodal LLM (e.g. MiniGPT-4 [38], mPLUG-owl [36]) capable of processing multimodal inputs. The purpose of this preprocessing is to summarize the features, reducing redundancy and preserving long-term contexts. Formally, this process can be defined as follows:

$$x_i = \left[ LLM_g(t_{h_i}), LLM_g(v_{h_i}), \cdots \right],$$
$$y_i = \left[ LLM_g(t_{c_i}), LLM_g(v_{c_i}), \cdots \right],$$

where $t, v, \cdots$ are textual, visual and other multimodal features, $x_i$ and $y_i$ denote the summarized data of historical items and conversations. $LLM_g$ represents the generating operation of LLM, distinguishing from its forward operation $LLM_f$.

*3.2.2* ***Construction of prompt.*** Using the behavior information $\mathbf{x}, \mathbf{y}$, we can construct a prompt to extract user preferences with the help of the LLM. There are three additional components: the instruction principle $p$, attribute $a_i$, and examples $e$. These components are artificially designed for each scene. The principle $p$ describes the task being performed by the LLM, which is "user preference extraction". The attributes $\mathbf{a}$ are tailored for each scene, such as "color, material, shape" for clothes or "genre, director, origin" for movies. In each question, LLM is assigned the task of answering user preferences related to a specific attribute, and the answers are later combined. The examples $e$, which provide the desired output format and example keywords (e.g., "cute", "cartoon", etc.), not only assist in guiding the LLM's responses but also follow a standardized output format, thereby facilitating the extraction of keywords from the generated output. Using this prompt, we can represent the keywords $\mathbf{k}_i^p$ generated by LLM for attribute $a_i$ as follows:

$$\mathbf{k}_i^p = LLM_g\left(p, a_i, e, \mathbf{x}, \mathbf{y}\right).$$

Next, we combine the outputs of each attribute and eliminate any duplicates to obtain preference keywords $\mathbf{k}^p$. The process of generating the target item keywords $\mathbf{k}^t$ is similar but with only one target item $h^t$ and its corresponding summarized information $x^t$. In this case, there are no conversations involved, and there is only a single attribute to consider:

$$\mathbf{k}^t = LLM_g\left(p, e, x^t\right).$$

### 3.3 Generate Soft Preference Embeddings

We have developed a method that relies solely on explicit keywords for representation. However, natural language, as a discretized form, has limited expressive capabilities with limited length. On the other hand, utilizing continuous hidden embeddings, which offer more informative and precise representations, requires substantial training resources. We utilize natural language as the baseline while training soft preference embeddings as an extra signal to correct this language bias with the help of an LLM, named bias correction LLM. These embeddings assist in addressing the mismatch between the natural language baseline and the actual user interests. The model is illustrated in Figure 3.

*3.3.1* ***Bias Correction LLM..*** The primary objective of the LLM is to predict the next textual token so it can only understand and generate texts. However, when applied to multimodal generation, it becomes necessary to introduce multimodal tokens to acquire the ability of multimodal generation. We incorporate multimodal tokens as learnable parameters into the embedding table and then utilize a linear layer to align the embedding space of the LLM with that of the generator. This alignment ensures consistency and compatibility between the LLM and the text encoder of the generator, facilitating the generation process. Additionally, we employ P-Tuning V2 [20] to fine-tune the LLM specifically for the generation task, which can enhance its generation ability. During each inference, the multimodal tokens are appended after the user behavior prompt. The soft preference embeddings are obtained by passing these augmented inputs through the LLM (with P-Tuning V2) and the linear layer.

Formally, in conjunction with the user behavior prompt $p, \mathbf{x}, \mathbf{y}$ constructed in section 3.2, we include additional multimodal tokens

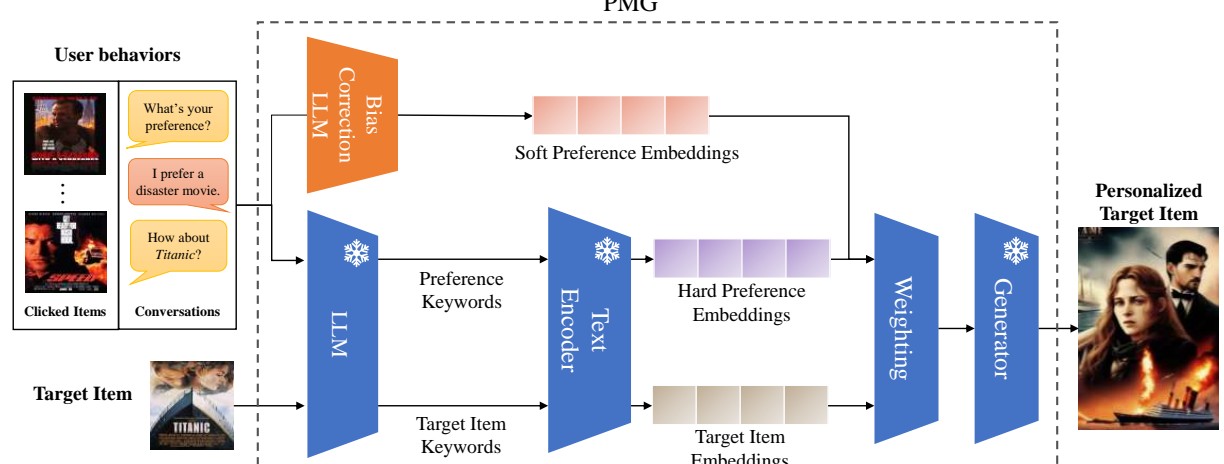

**Figure 2: Overview of our method. By utilizing user behaviors and a target item as input, we generate personalized multimodal responses for the item, taking a movie poster as an example in the figure.**

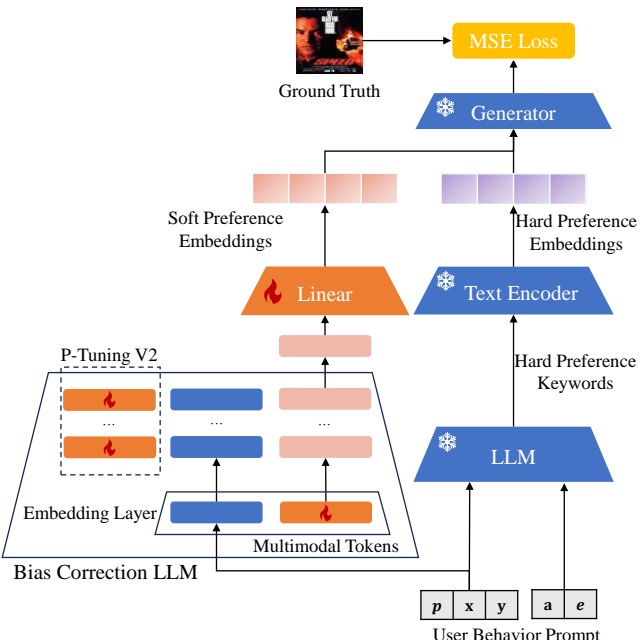

**Figure 3: Model designed to train soft preference embeddings.**

$\mathbf{m} = \{m_1, \cdots, m_L\}$ of length $L$. Attributes and examples are not utilized in this context, as the prefix embeddings have the ability to learn them on their own. These tokens are passed to the LLM, and their corresponding embeddings in the embedding layer are trainable. Following the P-Tuning V2 approach, $S$ trainable prefix embeddings $\mathbf{t} = \{t_1, \cdots, t_S\}$ are prepended to the embedding sequence in the self-attention of each transformer layer. The resulting output embeddings in the LLM's forward operation can be

represented as:

$$\text{prompt} = (p, \mathbf{x}, \mathbf{y}),$$

$$\left[\mathbf{E}_{prompt}, \mathbf{E}_m\right] = LLM_f\left(\mathbf{t}, \text{prompt}, \mathbf{m}\right),$$

where $\mathbf{E}_{prompt}, \mathbf{E}_m$ represent the output embedding of LLM, and the soft preference embeddings $\mathbf{E}_m$ is used for the subsequent multimodal generation process.

*3.3.2* ***Training with multimodal supervision.*** In contrast to GILL [18], which solely relies on captions for supervision, we believe that incorporating multimodal supervision (such as real images or audios) is more meaningful and helps to correct deviations. However, this approach introduces the challenge of propagating gradients backward through the generator, resulting in increased training difficulty. To simplify training, we utilize the preference keywords generated in section 3.2 as a foundational framework and focus on training a limited number of soft preference embeddings as additional conditions for the generation process.

The preference keywords are tokenized and transformed into hard preference embedding $\mathbf{E}_k$ by the text encoder of the generator. Then, we concatenate the $\mathbf{E}_m$ and $\mathbf{E}_k$ as conditioning input for the generator. Regarding data splitting, since it is impossible to obtain a real personalized image as ground truth, we use the last item in the interaction sequence as supervision and the previous ones as input.

Different generator models have different training algorithms. In our implementation, we utilize a diffusion model, which contains a text encoder and a U-Net [27]. The U-Net is employed as a conditional denoising module to generate images through multiple denoising steps. Following its training process, we introduce random noise $\epsilon \sim \mathcal{N}(0, 1)$ to the multimodal supervision $M_s$ and then attempt to denoise it:

$$\mathbf{E}^p = concatenate(\mathbf{E}_m, \mathbf{E}_k)$$

$$M_n = M_s + \epsilon,$$

$$M_d = Unet(\mathbf{E}^p, M_n).$$

The loss is calculated as MSE loss of $M_s$ and $M_d$:

$$loss = MSE(M_s, M_d).$$

Using this loss, we train the embeddings of multimodal tokens, and prefix embeddings in P-Tuning v2 to enable the multimodal generation ability of LLM, together with the mapper layer to align embedding space.

## 3.4 Balancing the accuracy score and the preference score

Different from the training process of soft preference embeddings including only preference conditions, the generation inference process incorporates both preference and target item conditions. Simply combining these conditions can result in favoritism towards one and overshadowing the other. Following previous studies such as DreamBooth [28] and GILL [18], we use the similarity between the generated responses and the preference keywords to measure the degree of personalization, which we call the *preference score*, and the *accuracy score* refers to the similarity with the target item keywords. The accuracy score measures the level of consistency with the target item, while the preference score about preference conditions gauges the degree of personalization. To balance them, we employ a weighted sum of accuracy score and preference score using pre-trained multimodal networks (e.g., CLIP [25], CLAP [5]).

Assuming the multimodal response $M$ is generated by the generator:

$$M = Generator(w_p \cdot \mathbf{E}^p, w_t \cdot \mathbf{E}^t),$$

where $w_p$ and $w_t$ are weights of preference and target item conditions to be adjusted. Through the encoders of the pre-trained multimodal network, we can transform the response $M$ and keywords $\mathbf{k}^p, \mathbf{k}^t$ into embeddings $e_M, e_p, e_t$. Then we can calculate the cosine similarity between them as the preference score $d_p$ and accuracy score $d_t$.

$$d_p = \frac{e_M \cdot e_p}{\|e_M\|_2 \|e_p\|_2},$$

$$d_t = \frac{e_M \cdot e_t}{\|e_M\|_2 \|e_t\|_2}.$$

Finally, our objective is to optimize the weighted sum of $d_p$ and $d_t$.

$$z = \alpha \cdot \log d_p + (1 - \alpha) \cdot \log d_t.$$

The hyper-parameter $\alpha$ is normally 0.5 and can be adjusted to achieve different effects according to usage scenarios and needs.

Considering the powerful parallel generation ability of current multimodal generators, we generate the response with multiple predefined sets of weights $w_p, w_t$ and pick the response with the highest score $z$.

## 4 EXPERIMENT

Our method can be used to generate various multimodal responses, encompassing not only images and audios but also other modalities. In this section, we focus on the generation of images as it is considered the most common and intuitive modality. Our experiments aim to answer the following research questions:

- **RQ1**: Can PMG accurately generate images that combine user preferences?
- **RQ2**: Why is conditions weighting necessary?
- **RQ3**: How do explicit keywords and implicit embeddings impact performance?
- **RQ4**: Are P-Tuning v2 and multimodal tokens beneficial while training soft preference embeddings?
- **RQ5**: Are there any additional purposes or applications for the generated images beyond user display?

### 4.1 Experimental Setup

*4.1.1 Scenarios and dataset.* We design the following three scenarios to verify our method:

(1) **Generating personalized images** of products whose original images are missing according to the historically clicked products of the user. We adopt POG [2], a multimodal dataset of fashion clothes, for training and evaluation. We selected 2,000 users and 16,100 items for experiments.

(2) **Generating personalized posters of movies** according to historical watched movies of user. We adopt the small version of MovieLens Latest Datasets [13], which contains 9,000 movies, 600 users, and 100,000 rating interactions.

(3) **Generating emoticons** in instant messaging according to current conversation and historically used emoticons of the user. Since we cannot find a suitable dataset, in scenarios we do not train soft preference embeddings and only use keywords to generate images.

*4.1.2 Implement details.* In all of our experiments, we select Llama2-7B [29] as the basic LLM model and Stable Diffusion V1.5 [26] as the image generator. Due to limitations of the dataset, at most $n = 10$ historical items and only the current $m = 1$ conversation are considered in the prompt of user preferences extraction. Then 10 personalized keywords and 5 target keywords are extracted for image generation. In the training of soft preference embeddings, $L = 4$ multimodal tokens and $S = 4$ prefix embeddings are used to get personal embedding. In our experiments, this training process costs 12 hours on a single NVIDIA V100 GPU. As for inference, each image costs about 5 seconds, 2s for LLM and 3s for stable diffusion.

*4.1.3 Evaluation metrics.* We employ multiple image similarity metrics to assess the resemblance between the generated image and historical/target items, quantifying the level of visual personalization achieved. To prevent potential information leakage, we exclude the CLIP metric used in the weighting module from this evaluation. Instead, we utilize the following two metrics:

(1) **LPIPS** (Learned Perceptual Image Patch Similarity) [37]: This metric measures the perceptual similarity between two images by considering human visual perception. It focuses on capturing semantic information.

(2) **SSIM** (Structural Similarity Index Measure) [32]: Widely used in image similarity assessment, this metric considers luminance, contrast, and structural information. It places more emphasis on image quality.

By employing these metrics, we can comprehensively evaluate the visual similarity between the generated image and the historical/target items, providing insights into the effectiveness of our personalized generation approach.

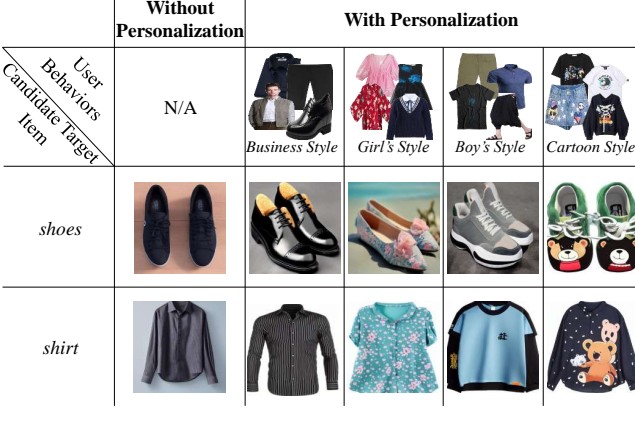

| | Without Personalization | With Personalization | | | |
|---|---|---|---|---|---|
| User Behaviors / Candidate Target Item | N/A | Business Style | Girl's Style | Boy's Style | Cartoon Style |
| shoes | | | | | |
| shirt | | | | | |

**Figure 4: Generated image comparison of our method PMG in the costume scene. Four typical users with different styles of historical items are picked as input to generate images of shoes and a shirt.**

| | Without Personalization | With Personalization | | |
|---|---|---|---|---|
| User Behaviors / Candidate Target Movie | N/A | Cartoon Lover | Thriller Lover | Romance Lover |
| True Crime | | | | |
| Titanic | | | | |

**Figure 5: Generated image comparison of our method PMG in the movie poster scene. Three users with different movie interests are picked as input to generate posters of movie *True Crime* and *Titanic*.**

## 4.2 Image Comparison (RQ1)

In this section, we show the generated images in three scenes: the costume scene, the movie poster scene and the emoticon scene. The existing personalization generation methods such as Textual Inversion [6] and DreamBooth [28] train extra embeddings for each user using their historical item images. They are only suitable for scenarios with a small number of users as they can consume significant training resources. As a result, they are not used in our experiments as baselines.

In the costume scene (Figure 4), PMG demonstrates notable personalization capabilities, particularly in cartoon and girl's styles. In the cartoon style, PMG identifies the association of these items with a specific cartoon character and accordingly selects a cartoon bear as the generated output. In the girl's style, PMG incorporates numerous floral patterns that align with girls' preferences.

| | Without Personalization | With Personalization | | |
|---|---|---|---|---|
| User Behaviors / Current Conversation | N/A | Animal | Computer | Sport |
| Support / *Don't worry!* | | | | |
| Dismayed / *I have no words.* | | | | |
| Happy / *Great! We won!* | | | | |
| Tired / *I'm running on empty.* | | | | |
| Sad / *My laptop is broken.* | | | | |

**Figure 6: Generated image of our method PMG in the emoticon scene. We generated emoticons for three users who have used different types of emoticons in five different emotional conversations. The current conversation serves as the target item.**

In the movie poster scene (Figure 5), PMG adeptly combines user preferences with the target item. For instance, in the case of the thriller movie *True Crime*, PMG consistently incorporates crime and horror elements into the generated posters, regardless of the user generating them. In the case of the romance movie *Titanic*, the generated posters consistently feature a couple in love, while the styles vary based on user preferences. Notably, for the cartoon enthusiast, both posters are transformed into cartoon drawings to align with their preference.

In the emoticon scene (Figure 6), we generate emoticons based on the ongoing conversation and previously used emoticons. Utilizing historical emoticons, the LLM (Large Language Model) helps summarize the user's preferences and designs a cartoon character, such as a cat or a boy with a football. By analyzing the conversation, the LLM assists in identifying the user's current emotion and devises a suitable pose for the emoticon, such as crying sadly or squinting from fatigue. The character and the pose can be considered as the personalization condition and target condition respectively, in order to generate the final emoticon. In the case without personalization, the LLM randomly designs a generic character.

As a result, we generate emoticons featuring a cat for animal lovers and emoticons relating to balls for sports enthusiasts, among others, and the emotions conveyed are generally accurate. However, the tears in the last conversation's emoticon are not rendered well. The main reason is the lack of an emoticon dataset to fine-tune the soft preference embedding condition, resulting in subpar performance when it comes to certain details. Furthermore, when generating animal-themed emoticons, our method consistently produces a cat since two of four historical emoticons depict a cat. Enhancing diversity when designing characters using large language models poses a challenge for our future work.

### 4.3 Case study of weighting (RQ2)

As explained in Section 3.4, directly combining personalization and target conditions can result in an imbalance. In Figure 7, we observe variations in the generated poster while adjusting condition weights for a romantic target movie *Titanic* and a disaster enthusiast. When the condition weights are set to $w_p : w_t = 0 : 4$, the poster predominantly considers the target condition (romance) and depicts a couple in love. Conversely, when the weights are adjusted to $w_p : w_t = 4 : 0$, the poster focuses solely on the personalization condition (disaster) and portrays a ship in a storm.

In order to incorporate both romance and disaster while following our selection principle outlined in Equation 1, we evaluate the generated posters based on their $z$ scores. Among the options, Figure 7b achieves the highest $z$ score and is selected as the final output.

### 4.4 Ablation Study

*4.4.1 Personalization conditions. (RQ3)*. In this section, we examine the contribution of the two forms of user preference representation, preference keywords, and soft preference embeddings (Table 1). By calculating the similarity between generated images and historical items, we can measure the degree of personalization, and by calculating the similarity with the target item, we can ensure that our generation does not deviate from the target.

Although our method focuses on introducing user preferences (reflected in historical items), the similarity with the target item did not decrease and even increased in movie scenes. This means personalization can also smooth out the errors between the generator and the real scene. As the backbone of our method, the keywords can greatly enhance the similarity of generated images in both LPIPS and SSIM metrics. However, the soft preference embeddings help to reduce the LPIPS metric but have no significant impact on the SSIM metric. This suggests that embeddings introduce more personalized semantic information but do not contribute to image quality due to their instability. When we combine both preference keywords and soft preference embeddings, we achieve the richest personalized content without deviating from the target items, while also ensuring the quality of the generated images.

Figure 8 is a case study on the soft preference embeddings. When provided with only the keywords "shoes, cartoon", there is a certain probability of generating cartoon-style drawings of shoes. However, after incorporating the hard preference embedding, the model consistently generates realistic shoes adorned with cartoon patterns.

*4.4.2 Prompt tuning. (RQ4)*. In this section, we analyze the impact of P-tuning V2 and multimodal tokens on the degree of personalization, measured by LPIPS similarity between generated images and historical items. Table 2 showcases their effectiveness. P-tuning V2 greatly enhances the ability of LLM to extract user preferences. Similarly, multimodal tokens exhibit a positive effect, although they also occupy a limited condition embedding and reduce the number of effective keywords. Therefore, the number of multimodal tokens should not be large, and setting $L = 4$ or $L = 8$ is determined to be the optimal parameter.

### 4.5 Auxiliary Generation (RQ5)

Our approach extensively explores interest modeling with LLM, enabling the generation of images that can be utilized not only for displaying to users but also for downstream recommendation tasks. This section presents an experiment conducted on the MovieLens dataset, aiming to evaluate the impact of incorporating generated images as additional visual features. To perform the evaluation, we employ MMGCN [34] as the base recommendation model, which focuses on leveraging multi-modal features to enhance recommendation performance.

The MovieLens dataset inherently includes image features of items, specifically the original movie posters, but it lacks image features for users. As a result, we have designed the following experiments: (1) **No-image**: This experiment does not utilize any image features and relies solely on the IDs of items and users. (2) **Item-only**: This experiment solely utilizes the image features of items. (3) **Averaged-user**: In addition to item image features, user image features are initialized as the average of historically watched items. (4) **Generated-user**: In addition to item image features, user image features are initialized as the image generated by PMG. It is important to note that the generated images are created under the personalization conditions, without a target item.

Table 3 provides compelling evidence that the inclusion of image features for items or users significantly enhances recommendation accuracy. Notably, incorporating the images generated by PMG yields superior results compared to the simple average baseline. These findings underscore the effectiveness of our approach in capturing user interests by leveraging the reasoning capability of LLM. By incorporating the generated images, our method successfully captures and incorporates nuanced user preferences, leading to improved recommendation performance.

One limitation of this experiment is the utilization of the small version of the MovieLens Latest Datasets, similar to the previous experiment. This dataset scale is comparatively smaller compared to the experiments conducted in the MMGCN paper. The primary constraint we faced was the time-consuming process of generating images using LLM and stable diffusion. We aim to address this limitation in future work.

## 5 CONCLUSION AND FURTHER WORK

In this paper, we have proposed a method named PMG for personalized multimodal response generation using LLMs. By leveraging large language models, we extracted user preferences and used them to condition the generation process of a generator. The experiments on image generation validate the effectiveness of PMG

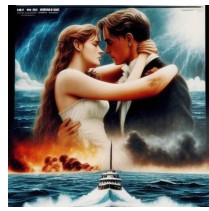 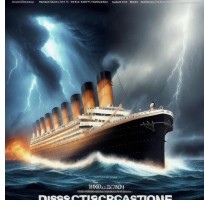 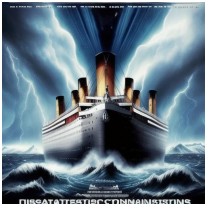 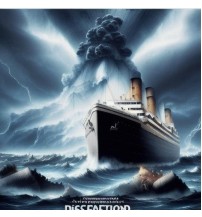

| (a) $w_p : w_t = 0 : 4$ | (b) $w_p : w_t = 1 : 3$ | (c) $w_p : w_t = 2 : 2$ | (d) $w_p : w_t = 3 : 1$ | (e) $w_p : w_t = 4 : 0$ |
|---|---|---|---|---|

**Figure 7: Generated poster of movie *Titanic* with different weights of conditions. $w_p$ is the weight of personalization conditions, which prefer disaster movie. $w_t$ is the weight of target item conditions, which consider it as a romantic movie. When $w_p : w_t = 1 : 3$ it achieves the highest $z$ score and the generated poster is a combination of romance and disaster.**

**Table 1: Quantitative ablation study of keywords and soft embeddings of personalization conditions. The best results are in bold and the second-best results are underlined.**

| Dataset | POG | | | | MovieLens | | | |
|---|---|---|---|---|---|---|---|---|
| Metric | LPIPS(↓) | | SSIM(↑) | | LPIPS(↓) | | SSIM(↑) | |
| | History | Target | History | Target | History | Target | History | Target |
| PMG | **0.5375** | **0.5482** | 0.1640 | 0.1600 | **0.4190** | **0.4140** | 0.2486 | **0.2515** |
| w/o embeddings | 0.5455 | 0.5592 | **0.1652** | **0.1608** | 0.4215 | 0.4176 | **0.2488** | 0.2505 |
| w/o keywords | 0.5616 | 0.5535 | 0.1533 | 0.1590 | 0.4406 | 0.4390 | 0.1867 | 0.1858 |
| w/o both | 0.5626 | 0.5526 | 0.1531 | 0.1567 | 0.4561 | 0.4542 | 0.1589 | 0.1575 |

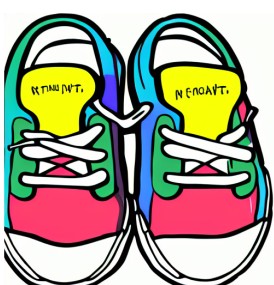 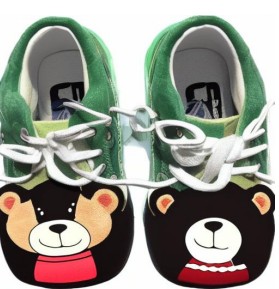

| (a) W/o soft embeddings. | (b) W/ soft embeddings. |
|---|---|

**Figure 8: A case study of personalization soft preference embeddings reveals the presence of language bias. With keywords "shoes" and "cartoon", the generation without these embeddings may produce a cartoon-style drawing of shoes. The generation with them consistently generates realistic shoes adorned with cartoon patterns.**

and its potential for downstream recommendation tasks. This work paves the way for further advancements in personalized generation, enabling the creation of tailored and engaging user experiences.

In future work, we plan to improve the diversity of the generated response. For example, the generated emoticons in Section 4.2 may sometimes be too similar to each other. Additionally, both LLMs and diffusion models require lots of time for reasoning, so we may investigate ways to improve the efficiency of the whole process.

**Table 2: Quantitative ablation study of P-tuning V2 and multi-modal tokens using the LPIPS metric on the POG and Movie-Lens datasets. $L$ denotes the number of multimodal tokens. The best results are in bold and the second-best results are underlined.**

| ID | P-Tuning V2 | $L$ | POG | MovieLens |
|---|---|---|---|---|
| 1 | ✗ | 2 | 0.4398 | 0.5471 |
| 2 | ✗ | 4 | 0.4353 | 0.5522 |
| 3 | ✗ | 8 | 0.4421 | 0.5586 |
| 4 | ✗ | 16 | 0.4482 | 0.5690 |
| 5 | ✓ | 2 | 0.4230 | 0.5453 |
| 6 | ✓ | 4 | 0.4190 | **0.5375** |
| 7 | ✓ | 8 | **0.4155** | 0.5386 |
| 8 | ✓ | 16 | 0.4212 | 0.5406 |

**Table 3: Comparison of the recommendation performances between MMGCN leveraging different image features of items and users. The best results are in bold and the second-best results are underlined.**

| | Item | User | Recall@10 | NDCG@10 |
|---|---|---|---|---|
| No-image | ✗ | ✗ | 17.57% | 0.0859 |
| Item-only | ✓ | ✗ | 18.88% | 0.0947 |
| Averaged-user | ✓ | Average | 19.54% | 0.0989 |
| Generated-user | ✓ | Generated | **20.03%** | **0.1004** |

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
