# OpenReview forum: "PMG : Personalized Multimodal Response Generation with Large Language Models"
_ACM.org/TheWebConf/2024/Conference — TheWebConf24_

### Official Review · Reviewer_dwGL · 2023-11-14

**Novelty:** 6
**Technical Quality:** 5

**Review:**

The paper introduces a personalized multimodal generation model that learns user preferences by analyzing their historical behaviors across various modalities. It generates both explicit keywords and implicit embeddings to represent user preferences. The model's response generation is trained using a weighted sum that blends the similarity to user preference keywords (measured by a preference score) with the similarity to target item keywords (measured by an accuracy score). Subsequently, a diffusion model is utilized for the actual generation of images.

Pros:
1. The paper proposes a novel and intriguing idea for personalized multimodal generation. The focus on personalized content creation within recommendation systems is both innovative and interesting. It is likely to draw attention to both the generative AI and recommendation communities.
2. The proposed algorithm is straightforward and well-explained within the paper.
3. The case studies showcasing the generated images reveal interesting and promising results. Although lacking quantitative baselines for comparison, this paper has the potential to serve as a benchmark for future research.

Cons:
1. All experiments are conducted on a similar image generation task, thereby not demonstrating the generalizability of the proposed multimodal generation model to other data sources.
2. Given the crucial role of generating explicit keywords from multimodal user history in this method, the paper would benefit from including case studies focused on these generated keywords. However, they are not presented in the paper.
3. Some details in the experimental setups remain unclear, such as the division of datasets into training and testing sets. Moreover, the reason behind certain choices, like using only 1 conversation from the history and generating 10 personalized keywords along with 5 target keywords, is not explicitly clarified.
4. A potential concern with the proposed algorithm is its efficiency. In certain scenarios, like emoticon generation, a 5-second response time may render the method impractical.

**Questions:**

The paper is generally interesting, but I have concerns about potential issues of image hallucination. For instance, in the poster generation scenario, the generated posters may not accurately relate to the movies, and the depicted characters could be fake, not reflecting real actors or actresses. Have you considered this issue in your research?

**Reviewer Confidence:**

3: The reviewer is confident but not certain that the evaluation is correct

**Scope:**

3: The work is somewhat relevant to the Web and to the track, and is of narrow interest to a sub-community

---

### Official Review · Reviewer_TVDQ · 2023-11-26

**Novelty:** 4
**Technical Quality:** 3

**Review:**

The paper introduces PMG (Personalized Multimodal Generator), a novel method for generating personalized multimodal responses using large language models (LLMs). It focuses on converting user behaviors into natural language, which then facilitates LLMs in extracting user preferences. These preferences are used to generate personalized responses in various forms, like text and images. PMG uniquely combines explicit keywords and implicit embeddings to represent user preferences, optimizing a balance between accuracy and preference scores in the generated responses.

- **Pros:**
  1. The first method uses LLMs for generating personalized multimodal responses.
  2. The paper is well-organized and easy to follow.

- **Cons:**
  1. **Lack of Detail in Methodology and Experiments**: The paper does not provide sufficient details in its method and experimental sections, making reproducibility challenging. Please check the Questions part.
  2. **Misalignment with Practical Recommendation Scenarios**: The approach essentially performs style transfer on target item images to align with user preferences, rather than generating images of the target items themselves. This is less relevant to real-world recommendation scenarios, such as in the provided movie example where the generated movie posters reflect style preferences but have incorrect titles, which could be confusing for users.
  3. **Need for Comparison with Similar Techniques**: The paper's approach closely resembles image style transfer techniques. A comparison with other methods in this field would enhance the paper's context and relevance.

The paper is well-written overall. However, it could benefit from a more detailed explanation of data preprocessing steps and examples, particularly in the sections describing the generation of explicit keywords and soft preference embeddings.

**Questions:**

1. Please further explain the limitation mentioned in Section 2.2 regarding VIP5's reliance on original item images and how PMG addresses this issue.
2. In Section 3.2, the paper describes using LLMs to convert user behaviors into text descriptions and then extracting keywords. Can you provide specific examples of prompts used in these steps, how they were designed for different scenarios, and whether including more attributes improves the outcome?
3. Why is only one attribute used for the target item and why is it inconsistent with preference keywords?
4. How are multi-modal tokens generated? Can you provide an example?
5. Do the clothes and movielen datasets both include dialogue? How were they constructed?
6. The implementation details section lacks critical information such as learning rates, batch sizes, the deep learning framework used, and training epochs. Are there any differences in the experimental setup between the two datasets? Please clarify these details.

**Reviewer Confidence:**

4: The reviewer is certain that the evaluation is correct and very familiar with the relevant literature

**Scope:**

3: The work is somewhat relevant to the Web and to the track, and is of narrow interest to a sub-community

---

### Official Review · Reviewer_66mU · 2023-12-06

**Novelty:** 5
**Technical Quality:** 3

**Review:**

The paper under review tackles a problem of personalized multimodal generation. In other words, given a set of previous items, the model can generate a target image that is personalized to these items.

This is indeed a novel problem statement, and I haven't seen this in the literature before. The architectures themselves that are employed are fairly standard, and rely on a concatenation of text and multimodal embeddings to generate a personalized image from a target image. To personalize, the model is trained on user behavior signals (e.g., clicks), such that the generated image should better reflect the user preferences.

**Questions:**

1) I have several concerns about the evaluation. First, I am not sure that LPIPS and SSIM are appropriate metrics here. All we do, to the best of my understanding, is to verify that there is a visual similarity between the generated image and the prior user items. This is a somewhat "self-fulfilling prophecy", as this similarity is exactly what the model is trained for. Instead, it would be good to have some sort of human evaluation of item personalization. Are users more satisfied with non-personalized original items, or the personalized ones?

2) In addition, the generated item should make sense in the context of the target item. E.g., how do we know that the poster generated for the Titanic movie, actually reflects the content of the movie? I am worried that the authors completely disregard the dangers of hallucination in their approach.

3) The evaluation on the third dataset is unclear. If the soft preferences are not trained, how is the proposed technique in fact different from just concatenating the current conversation with prior keywords from the user? E.g., looking at figure 6, first row. Is the generated image just coming from "Don't worry + Animal / Computer / Sports" keywords?

**Reviewer Confidence:**

3: The reviewer is confident but not certain that the evaluation is correct

**Scope:**

3: The work is somewhat relevant to the Web and to the track, and is of narrow interest to a sub-community

---

### Official Review · Reviewer_bbKD · 2023-12-07

**Novelty:** 5
**Technical Quality:** 6

**Review:**

This work presents a method to personalize image response generation based on user behavior logs. The main idea is to use LLMs to extract user preferences to guide the image generation process. It also propose a few techniques including
- Augment the extracted keywords (from LLMs) with soft embeddings to represent the user preference.
- Control the bias toward user preferences using a weighted scoring function (Sec 3.4).

Strength
- Personalized generation is an interesting topic
- The method proposed is technically sound – reasonable solutions to a few technical challenges.
- Experiments are relatively comprehensive.

Weakness
- Novelty is not high. That said, the work provides good technical contributions in terms of solving practical challenges in developing personalized image generation.

**Questions:**

n/a

**Reviewer Confidence:**

1: The reviewer's evaluation is an educated guess

**Scope:**

4: The work is relevant to the Web and to the track, and is of broad interest to the community

---

### Decision · Program_Chairs · 2024-01-22

**Decision:**

Accept

**Comment:**

The paper presents a personalized multimodal response generation framework for web applications, including generating personalized images, generating personalized posters of movies, and generating emoticons. Generating personalized contents is an interesting direction that is worth exploration, which helps to tailor the generative models for different user preferences. The proposed method fills the gap with three example applications. Authors are advised to discuss the hallucination problem in personalized generation, and include some multimodal recommendation baselines such as VIP5 for comparison of the recommendation accuracy.